# A Review on Nanocellulose and Superhydrophobic Features for Advanced Water Treatment

**DOI:** 10.3390/polym14122343

**Published:** 2022-06-09

**Authors:** Danish Iqbal, Yintao Zhao, Renhai Zhao, Stephen J. Russell, Xin Ning

**Affiliations:** 1Shandong Center for Engineered Nonwovens, Industrial Research Institute of Nonwovens & Technical Textiles, College of Textiles & Clothing, Qingdao University, Qingdao 266071, China; danish.iqbal@ymail.com (D.I.); 17864222853@163.com (Y.Z.); chinesezrh@126.com (R.Z.); 2Leeds Institute of Textiles and Colour (LITAC), School of Design, University of Leeds, Leeds LS2 9JT, UK; s.j.russell@leeds.ac.uk

**Keywords:** nanocellulose, water treatment, superhydrophobic coating, Janus membrane, membrane technology

## Abstract

Globally, developing countries require access to safe drinking water to support human health and facilitate long-term sustainable development, in which waste management and control are critical tasks. As the most plentiful, renewable biopolymer on earth, cellulose has significant utility in the delivery of potable water for human consumption. Herein, recent developments in the application of nanoscale cellulose and cellulose derivatives for water treatment are reviewed, with reference to the properties and structure of the material. The potential application of nanocellulose as a primary component for water treatment is linked to its high aspect ratio, high surface area, and the high number of hydroxyl groups available for molecular interaction with heavy metals, dyes, oil-water separation, and other chemical impurities. The ability of superhydrophobic nanocellulose-based textiles as functional fabrics is particularly acknowledged as designed structures for advanced water treatment systems. This review covers the adsorption of heavy metals and chemical impurities like dyes, oil-water separation, as well as nanocellulose and nanostructured derivative membranes, and superhydrophobic coatings, suitable for adsorbing chemical and biological pollutants, including microorganisms.

## 1. Introduction

Water and land pollution have become a severe problem as a result of the industrial revolution, population growth, and rapid urbanization [1]. Pollutants such as heavy metals, saturated salts, oil emulsions, organic compounds, colors and dyes, and even microbes are found in wastewater. Cellulose nanomaterials offer a lot of potential in terms of wastewater treatment and management [2]. Different cost-effective adsorbents and superhydrophobic surfaces are now extensively utilized strategies for treating wastewater [3]. Economically viable water filtration and purification technologies are vital to support the continuing population growth and the improvement in standards of living across the developing world. At the start of the century, more than a billion people lacked access to safe drinking water and sanitation, and about 4000 children under the age of five die every day as a result of the “clean water crisis” [4]. Collaboration work by the United Nations International Children’s Emergency Fund (UNICEF) and World Health Organization (WHO) has resulted in significantly increased access to safe drinking water, from 82% in 2000 to 91% of the global population in 2015 [5]. Notwithstanding this outstanding achievement over a comparatively short time period, up to 675 million people were still without access to clean water in 2015, and 2.4 billion people lacked appropriate hygiene facilities [6]. WHO/UNICEF and other development programs face the greatest challenges in the least developed nations of the world, as well as in rural regions of the developing world. About 16% of the world’s rural people lack access to clean, drinking water, and an even larger proportion lack sanitation [7]. In rural areas of Africa, waterborne illnesses such as Guinea worm, diarrhea, cholera, and parasitic infections, continue to endanger lives every day. Globally, the vectors of infection that cause diarrhea result in 2–2.5 million deaths per year [8]. Consequently, abundant, cost-effective water treatment materials and processes that can be locally sourced are increasingly important to support sustainable development. 

Nanocellulose, the most plentiful bio-polymer on earth, has been identified as a suitable material for the scalable production of cost-effective, sustainable water treatment media [9]. Cellulose can be converted into various nanocellulose formats by a variety of physio-chemical processes, to provide highly attractive properties for wastewater treatment applications. By extracting nanocellulose from plentiful locally available and affordable natural biomass, there is significant potential to scale availability and deliver cost-effective sustainable water treatment media. Superhydrophobic materials, that show superior water repulsion owing to their low surface energy composition and unique micro- and nanoscale roughness, seem to be of particular importance due to their numerous applications spanning from self-cleaning surfaces to microfluidics. The other reason for the comprehensive scientific research on superhydrophobicity was to gain a thorough consideration of wetting concepts on various kinds of superhydrophobic surfaces, like the intrinsic superhydrophobic surfaces of lotus leaves, in which the quick transportation of droplets of water yields a self-cleaning outcome, or petals, where water particles attach strongly to the surface. The enormous potential of superhydrophobic nanocellulose-based materials for smart solutions and functional textiles has been recognized, and nanocelluloses have emerged as the most frequently utilized substrates for superhydrophobic coatings [10]. Figure 1 summarizes scientific articles relating to the use of nanocellulose in the treatment of wastewater and highlights the growing value of this particular application.

This review seeks to analyze and evaluate this expanding field of research by investigating nanocellulose separation techniques, characterization, and improvements, together with the efficacy of environmental restoration for water purification. It pays special attention to the superhydrophobic nanocellulose-based materials for oil/water separation membranes, Janus switchable membranes, and filters for water treatment. Advanced methods for efficient wastewater treatment are also being explored, including the application of fibrous electrospun membranes, switchable membranes, filters, and membrane bioreactors.

## 2. Structure and Properties of Nanocellulose

Cellulose is a naturally occurring macromolecule comprising multiple glucose units (Figure 2). Cellulose is the most abundant carbohydrate on the globe, with a projected yearly output of 10^10^ tons. It may be derived from numerous renewable sources, including plants, wood, algae, and microorganisms, among other sources. It is possible to separate cellulose into nano-sized components, in the form of nanocellulose with dimensions of 100 nanometers or lower [11]. The extracted nanocellulose can be derived either from the amorphous or crystalline regions of cellulose, which, of course, influences mechanical properties. So, nanocellulose, therefore, differs in chemical and physical properties from cellulose, has a high surface area, outstanding mechanical strength, and a high level of functionality and adaptability [12].

As shown in Figure 3, nanocellulose can be categorized in terms of cellulose nanocrystals (CNC), cellulose nanofibers (CNF), and bacterial nanocellulose (BNC) [14]. In the following sections, each type of nanocellulose is considered separately.

### 2.1. Cellulose Nanofibers (CNFs) Structure and Properties

In general, mechanical processes, specifically grinding, high-pressure homogenization, micro-fluidization, and ultrasonication can be used to abstract CNFs from plant lignocellulosic resources. These separation methods are particularly energy-intensive due to the high power required to fibrillate the cellulose and enable extraction of the constituent small diameter, longitudinal fibrils, or CNFs [12,15]. Before the mechanical isolation process, pretreatments are required to remove hemicellulose and lignin. Chemical treatment processes, for example, 2,2,6,6-tetramethylpiperidine-1-oxyl radical (TEMPO) oxidation, enzyme pretreatment, and acid-alkaline pretreatments can reduce the energy requirements [15]. CNFs are generally smaller than 100 nm in diameter, and their length is of the order of micrometers. CNFs have a higher aspect ratio than CNCs and have a long-chain structure, large surface area, and an abundance of hydroxyl groups on the surface for chemical bonding with other polymers [16]. The fibrous nature of CNFs improves compatibility with the manufacturing processes used for making sheet-like membranes to be used as water treatment media.

### 2.2. Cellulose Nanocrystals (CNCs) Traits and Structure

CNCs are the crystalline parts of CNFs that remain after the amorphous area is removed by depolymerization or acid hydrolysis. The amorphous area is an unstructured region that can be dissolved by acid under certain conditions. Strong acids, for example, sulfuric acid (H_2_SO_4_), are frequently employed in hydrolysis, and the characteristics of CNCs are highly dependent on the hydrolysis conditions, which include temperature, duration of reaction, and acid concentration [16]. CNCs are typically rod-shaped or needle-shaped with a diameter of less than 150 nm and a length of a few hundred nanometers. In comparison with CNFs, CNCs have high strength, superior thermal stability, and a high degree of crystallinity [12].

### 2.3. Bacterial Nanocelluloses (BNCs) Traits and Structure

In contrast to CNFs and CNCs, bacteria generate BNCs during the polymerization of organic substrates (sugar, glycerol) during cell culture [16,17]. As such, BNC is the purest form of nanocellulose available, as it contains no components of lignocellulose biomass other than cellulose. BNCs are twisted ribbons and have a width of 20 to 100 nm and a length in micrometers. While BNC shares the same chemical composition as other forms of nanocellulose, it is purer, has an excellent water-holding capability, and crystallinity, which results in superior thermal and mechanical strength [12,17]. In contrast to CNCs and CNFs, BNC is synthesized by bacteria and contains neither lignin nor hemicellulose. In experiments, membranes produced with this material have been demonstrated to remove various pollutants from water, including arsenic, mercury, vanadium, liquid dyes, chromium, phenol, lead, and natural organic matter [18,19,20]. Table 1 provides the association of the properties of CNFs, CNCs, and BNCs.

## 3. Nanocellulose Sources

The major sources of nanocelluloses include agricultural deposits, wood and plants, feedstocks, algae, bacteria, and wildlife as elaborated in Table 2.

## 4. Nanocellulose Preparation

### 4.1. Acid Hydrolyzation

The cellulose molecule is made up of an amorphous and a crystallization zone, and the amorphous zone may be eliminated by acid hydrolysis to produce nanocellulose with a high crystallinity. Hydrochloric acid, sulfuric acid, phosphoric acid, formic acid, hydrobromic acid, and others are examples of hydrolyzed acids. Controlling variables such as acid content, temperature, and duration of the hydrolysis process may produce nanocelluloses of various sizes and crystallinities. Dai et al. performed a series of processing methods on pineapple peel, and a needle-like structure with a mean width of 15 ± 5 nm and a length of 189 ± 23 nm was produced with 64 percent sulfuric acid by hydrolyzation [49]. The acid hydrolysis process is versatile, and the resulting nanocellulose has a homogeneous particle size and has already reached industrial production in some countries. However, recovering the residue is difficult, and post-treatment is inconvenient. The cellulose structure may be damaged or even sulfonated during the hydrolysis process. As a result, cleaner and more productive approaches have caught the interest of researchers.

### 4.2. Oxidative Process

The oxidative agent 2,2,6,6-tetramethylpiperidine-1-oxyl (TEMPO) is a weak one. However, sodium hypochlorite may transform it into a nitrogen carbonyl cation (a powerful oxidizing agent) that can preferentially oxidize the C6 hydroxyl groups on the cellulose surface to form aldehyde and carboxyl groups. Chang et al. [50] were the first to chitin, oxidize starch, cellulose, chitosan, and other polysaccharides using the TEMPO-NaClO-NaBr system in 1996. The findings revealed a high reaction yield and efficiency, as well as a significant improvement in polysaccharide water solubility. The TEMPO oxidation process features moderate reaction conditions, is easy to use, and uses little energy. The TEMPO system, on the other hand, incorporates hypohalite, which may produce chlorine, which is exceedingly damaging to the environment.

### 4.3. Physical Process

Physical approaches mainly entail defibrillating cellulose fibers with high-pressure mechanical processing and then separating nanocellulose. On the other hand, the nanocellulose formed by mechanical force has a large size and an irregular distribution. As a result, raw materials are often processed prior to mechanical processing, such as enzyme pre-hydrolysis, acid pre-hydrolysis, oxidation treatment, alkali swelling, and so on. Dufresne and his team [51] used highly pressurized homogenization to break the cell wall and extract the nanocellulose from the refined sugar beet. The mechanical fibrillation procedure for the manufacture of cellulose nanofibers from two different soft- and hard-wood cellulose pulps was examined by Stelte and Sanadi [52]. The diameter distribution of soft- and hardwood nanofibers is in the 10–25 nm range. The microfibril concentrations were then molded into a film to produce robust cellulose microfibril films. High pressure and temperature altered the structure of cellulose physically and chemically throughout the physical process.

### 4.4. Ionic Liquid Process

This is a liquid salt solution comprised of an organic anion and an organic cation, or nonpolar. These have a low or almost non-existent vapor pressure, excellent thermal and chemical invariability, a good ability to dissolve in polar and non-polar solvents, quick restoration, and non-flammability, among other properties. As a novel kind of eco-friendly solvent system, it offers a broad variety of applicability in cellulose dissolution. By liquefying microcrystalline cellulose in 1-butyl-3-methylimidazolium hydrogen sulfate, Tan and colleagues created rod-shaped cellulose nanocrystals. As a consequence, as the reaction temperature increased, the diameter of the CNC shrank, and the ideal reaction temperature was 90 °C, at which the formed CNC was tiny and crystallinity was high [53].

### 4.5. Biological Process

A single filament fiber is made up of multiple nano-scale microfibers, and ultrafine filament fibers are interlinked to produce BNC, which has a framework with a thickness of 20–100 nm and is commonly made by growing microorganisms. Alcohols and low molecular weight sugars are the major sources of BNC. Rhizobium, Acetobacter, Pseudomounas, Sarcina, Azotobacter, and other bacteria are capable of creating BNC [54]. BNC grew in popularity when Brown and his team found that acetobacter xylinum might also generate bacterial nanocellulose in 1986 [55]. BNC is a unique nanomaterial with great stability, high crystallinity, homogeneous particle size, and a controlled structure. Varied bacterial strains, culture conditions, and culture procedures may be used to manufacture BNC with different chemical characteristics. Biosynthesis is a low-energy, pollution-free technology. However, it takes a long time, is expensive, and requires a rigorous preparatory procedure. As a result, further research and development are still necessary for practical manufacturing.

### 4.6. Electrospinning Method

The electrospinning process involves passing a cellulose solution of high concentration through a metal needle-shaped syringe and pressing it firmly under a high voltage field to produce nanocellulose [56], and the nanocellulose produced does have a high aspect ratio. Currently, there is only a little research on electrospinning by simply liquifying cellulose, since cellulose contains few solvents, but the heating rate of the solvent is high, and also with the salt concentration, the electrospinning process is challenging. Kim et al. [57] used electrospinning to make cellulose nanofibers with a width of 200–750 nm from dissolved cellulose in lithium chloride/N, N-dimethyl acetamide, and N-methylmorpholine oxide/water solvent systems, respectively, and considered the effects of spinning parameters on CNF. The surface deposited cellulose nanofiber web is then employed as a filter membrane.

## 5. Cellulose-Based Materials in Industrial-Scale Water Treatment

Cellulose-based membranes are industrially applied in reverse osmosis (RO) for desalination, lowering the cost of membranes in multiple countries from USD 5.4 million to USD 0.5–1.2 million in 2019 [58] as compared to conventional desalination processes. In one specific example, the Sulaibiya water treatment facility in Kuwait City, Kuwait lowered the pretreatment costs by 2–7% by implementing nanocellulose-based ultrafiltration [59].

Membrane bioreactors are used for wastewater treatment in countries such as Singapore [60] and China [61]. The economic viability of water treatment depends on the facility’s production capacity, membrane life, and quality of water supply. Much of the work on forward osmosis (FO) membranes has taken place in the USA and Denmark, driven by companies such as Hydro Technology Innovations (HTI). FO is a technique for separating water from dissolved solutes that utilizes a semipermeable membrane and the inherent energy of osmotic pressure. The osmotic pressure is needed to drive water across the membrane while simultaneously retaining all dissolved solutes on the other side. The FO method has been used by HTI, NASA, and the fracking industry in the USA, where the capacity to recover eighty percent of the water from drilling has been demonstrated [62]. Oasys Water (Cambridge, MA, USA) and Beijing Woteer (Beijing, China) partnered to recover a large amount of wastewater produced by the Zhongtian Hechuang Energy Co. (Ordos, Inner Mongolia, China) in 2017. Before the cooperation was established, the ClearFlo Membrane Brine Concentrator was used to treat polluted wastewater successfully at a volumetric rate of ca. 5760 m^3^/d. Numerous countries, including Japan, Oman, Denmark, and Korea, have previously used the FO-RO system for large-scale wastewater treatment. Industrial-scale manufacturing of wastewater treatment membranes composed of cellulose has also been installed in several countries [62], including Denmark (Aquaporin, Kongens Lyngby, Denmark), Japan (Nitto Denko, Osaka Japan), the USA (Oasys Water, Fluid Technology Solutions Inc., Hydration Technology Innovations, Cambridge, MA, USA), and Korea (Hydration Technology Innovations, Woongjin Chemical Co. Ltd., Samsung Co. Ltd., Seoul, Korea). Furthermore, Trevi System Inc. (Rohnert Park, CA, USA) has developed many small-scale facilities in the USA and the Middle East, utilizing cellulose acetate FO membrane technology [63].

## 6. Superhydrophobic Nanocellulose-Based Wastewater Treatment Materials

Research on nanocellulose-based superhydrophobic materials has advanced rapidly in recent years due to potential applications, especially in self-cleaning [64] and oil-water separation [65,66]. Template etching, vapor deposition, coating technique, electrochemical approach, electro-spinning, and self-assembled technique have all been used to make superhydrophobic surfaces [67,68,69,70,71,72]. All of the aforementioned superhydrophobic surfaces were created using a combination of micro/nanostructure and lower surface free energy. Despite the development of several substrates and modification techniques, there are still certain barriers to practical implementation due to the use of ecologically unfavorable fluorinated materials and complex manufacturing processes.

Heavy metals, dyes, pesticides, polycyclic aromatic hydrocarbons, chemicals, and biomolecular contaminants are significant impurities produced by industry with the potential to enter water sources by various means. Due to its intrinsic properties, nanocellulose has the potential to provide enhanced performance in the treatment of water. High surface area and tensile strength, the ability to modulate surface chemistry by grafting anionic or cationic surface groups, and hydrophilicity are valuable characteristics of nanocellulose for wastewater treatment [73,74]. Functionalized nanocellulose, nanocomposites, nanohybrids, and superhydrophobic membranes are known to be effective adsorbents for pollutants for a variety of purposes, including organic contaminant cleanup from drainage discharge; isolation of heavy metal ions; isolation of oil; dye removal from industrial wastewater; disinfection of contaminated water; and a decrease in density and chemical oxygen needs in wastewater. To separate combinations of water and petroleum, ultra-porous nanocellulose aerogels have been used [75].

CNCs’ nanopapers and nano-coatings are also applicable for the cleanup of crude oil spills in saltwater. CNCs exhibit excellent pH and salt stability, as well as in the presence of crude oil. Most commonly, oils and organic contaminants are extracted from water using functionalized CNFs. CNC-based nanocomposite membranes are good at preventing fouling because they are water-friendly, have holes, and have a smooth surface. Researchers are looking into CNC binary systems to make them even more effective [76].

Nanocellulose-based membrane technologies for water treatment are not widely established in developing countries. The nanocellulose properties make it an intriguing and cost-effective material for wastewater treatment, providing the significant potential to service a growing sector. The material can be deployed as an adsorbent, a flexible membrane, or as a hybrid construct for wastewater treatment.

### 6.1. Theory of Pollutant Separation across Hydrophobic Surfaces or Interfaces

Wettability of a medium or surface refers to its capability to generate interfacial adhesion forces such as adhesive and cohesive forces when in interaction with a liquid. The degree of wettability is determined by the abovementioned forces that have an influence on the contact angle (CA) between a liquid and a substrate (Figure 4). The CA then assesses if a substance or substrate is hydrophilic, hydrophobic, or a combination of the two. Recently, a majority of scientists have been attracted to the production of hydrophobic nanomaterials owing to the growing need by both academia and industry. The hydrophobicity of nano-composites is a key feature in determining their applicability. Hydrophobic materials that are antifouling, self-cleaning, water repellent, and have low friction are highly sought for industrial utilization. Chemical alteration or surface roughness may be used to modify hydrophobicity. Surface energy reduction through chemical changes and increased roughness must be regulated concurrently to obtain superhydrophobicity [77]. Superhydrophobic materials are created via rigorous chemical and physical processes. Adhesion of molecules with low surface energy, in particular fluorinated agents [78], organic hydrophobic chains [79], silanes [77], etc., is one kind of chemical alteration used to achieve hydrophobicity.

The water contact angle (WCA) is an essential factor to consider while researching hydrophobicity. A hydrophilic substance has a WCA of less than 90°, a hydrophobic medium has a WCA greater than 90°, and a superhydrophobic material has a WCA greater than 150° [80,81]. To calculate WCA, many relationships such as Young’s equation, Wenzel equation, and Cassie model have been created [80]. The wetting of an excellent, chemically homogenous, and smooth surface may be calculated using Young’s equation [82]:cos θ_Y_ = (γSV − γSL)/γLV(1)
where ‘θ_Y_’ is the liquid CA and ‘γSV’, ‘γSL’, and ‘γLV’ are the interfacial tensions between solid-vapor, solid-liquid, and liquid-vapor, respectively.

Actual surfaces are seldom perfectly smooth. However, if a surface seems macroscopically plain, it usually contains micro-, nano-, and molecular-scale roughness. The wetting of rough surfaces has received significant attention after Wenzel [83], and Cassie and Baxter [84] introduced rough surface wetting equations (Figure 5). Wenzel, and Cassie and Baxter describe two contrasting wetting states: one in which the surface is completely wetted by a liquid (the Wenzel state), and the other in which air is held in the substrate’s roughness and only the top parts of the surface get wetted (the Cassie state).

The wetting theories of Wenzel and Cassie–Baxter are quite effective in anticipating and interpreting the wetting aspects of rough surfaces. The entire wetting condition on a rough surface is defined by the Wenzel equation:cos θ_W_ = r cos θ_Y_(2)
where ‘r’ is the principal surface area divided by the projected surface area, ‘θ_Y_’ is the CA of Young on the smooth surface of that material, and ‘θ_W_’ is the Wenzel’s CA on the rough surface. The Cassie–Baxter equation may be adapted to predict the amount of partial wetting on a rough surface:cos θ_CB_ = r_f_ f cos θ_Y_ + f − 1(3)
where ‘θ_CB_’ is the CA of Cassie–Baxter on the rough surface, ‘r_f_’ is the observed wetted area divided by the intended wetted area, and ‘f’ is the proportion of the surface’s estimated area that has been wetted by the fluid.

Extensive scientific study on superhydrophobic surfaces has been motivated by a need for comprehensive knowledge of wetting mechanisms on many kinds of superhydrophobic surfaces. The superhydrophobicity of lotus, butterfly wings, and other plant leaves has been well researched and documented in nature [79,80]. Lotus leaves, having a mechanism to clean themselves, have an uneven array with two stages of roughness [79]. The lotus-effect, which refers to the superhydrophobic qualities of lotus, has inspired the development of various artificial hydrophobic materials. Water droplets move easily and clean themselves on naturally superhydrophobic surfaces such as lotus leaves and rose petals, in which water vapors strongly attach to the surface. This twofold roughness of the waxy surface of the lotus leaf is the cause of its superhydrophobicity and exceptionally low water adherence. On the lotus surface, water CA greater than 160° has been observed [85]. In such observations, water vapors remain in the Cassie state on the lotus layer, and their facile movement is due to the limited area of contact between the solid surface and droplets. Rolling droplets, such as rainwater, may readily capture polluting particles from plant leaves, maintaining a clean surface. The self-cleaning consequence, often known as the lotus effect, of self-cleaning bio mimetic from natural surfaces, is one of the most commonly proposed purposes of manufactured superhydrophobic surfaces (Figure 6).

Furthermore, the WCA, the sliding angle, is an essential characteristic of hydrophobicity. The tilt angle between the surface and the substrate droplet at which the droplet starts to slide off the surface is known as the sliding angle. Superhydrophobic materials have a sliding angle of less than 10 and that frequently defines the water-repellent quality [10,80].

### 6.2. Models in Exploratory Research

Nature-inspired nanomaterials have increasingly gained popularity owing to their unique interfacial wetting capabilities [87]. A material’s surface chemistry and structure may be tuned to be superhydrophilic, superhydrophobic, superoliophilic, or superoliophobic [88]. Influenced by the lotus leaf phenomena, hierarchically structured membranes with superhydrophobic and superoliophilic qualities might reject high surface tension water droplets while still being wetted by low surface tension oils [89]. This unique property gives the material the capacity to suddenly separate oil and water, opening the door to high-efficiency oil/water isolation. The separation rate and flux, two significant markers of separation degree, have always been our goal in the oil-water segregation method. The Hagen-Poiseuille equation may be used to represent the flow [90]:*J* = ξΠr_p_^2^ΔP/8 µL(4)
where ‘*J*’ denotes the flux of the membrane, ‘ξ’ stands for the porosity of the membrane, ‘r_p_’ is the pores radius, ‘ΔP’ denotes the pressure drop, ‘μ’ is the liquid viscosity, and ‘L’ denotes the mesh thickness.

To create a hydrophobic surface, paper, a primary cellulose product, was coated with tetraethyl orthosilicate liquid and tridecafluorooctyl triethoxysilane. The modified paper exhibited superhydrophobic qualities with a WCA of more than 170° and a sliding angle of less than 7° [10]. Instead of employing solely silica nanoparticles, fluorinated chemicals are often used to improve water repellency. Outdoor athletic textile applications have recently shown an interest in water-repellent, anti-stain, and self-cleaning fibers [91]. In response to the growing need for hydrophobic materials, scientists are inventing many techniques for the hydrophobic transformation of nanocellulose. Hydrophobic and superhydrophobic materials have limited practical applicability owing to the vulnerability of nanocelluloses to environmental deterioration [92]. Zhou et al. [93] have made a new microfibrillated cellulose aerogel having superhydrophobic characteristics to effectively isolate oil and water (Figure 7). In this research, salinization in an ethanol solution having methyltriethoxysilane was engaged to establish polyxyloxane groups on the surface of the aerogel. Before salinization, the aerogel was oxidized in order to generate hydroxyl groups that served as the mainstay of the salinization procedure. The polysiloxane groups are very water-repellent (superhydrophobic) and oil-absorbent (lipophilic) inside the aerogel’s porous structure. When submerged in oil-contaminated water, the resultant modified aerogel displayed oil selectivity of up to 159 g/g. After 30 cycles, experimental tests demonstrated a reusability volume of up to 92 g/g or around 58%. The cellulose nanocrystals’ exceptional mechanical properties, including wear resistance, strength, and stiffness, are due to a strong covalent bond at the cellulose interface and are most likely responsible for the recyclability, endurance, and effectiveness of oil-water separation.

Laitinen and his team [94] transformed waste cellulose fibers by a freeze-drying technique, consistent in an adsorption capacity of 142.9 (g/g) and a great oil absorption capability for more than 30 cycles. These superhydrophobic absorbents could be used to clean up chemical spills and oil because they are made from cheap raw materials, use sustainable resources, and have easy and ecologically friendly production methods.

Fan et al. [95] researched and created an alterable superhydrophobic/superoleophilic superhydrophilic/superoleophobic transformation of improved cellulose material. The smart or switchover superhydrophilic-superhydrophobic cellulose material was made by immersing the material in an aqueous medium of sodium hydroxide, urea, and distilled water for a predetermined time, then treating it with a zinc chloride aqueous solution to allow chlorine anions and zinc cations to be absorbed onto the cellulose fibers’ surface. The ions may enter the pores because the fibers have swollen. The ions were imprinted on the textile fibers by steaming the charged fabric. The treated material was then rinsed many times with deionized water before being baked. The baking process shrinks the fibers, substantially trapping the imprinted ions in the fiber spouts when water is withdrawn and shrinkage proceeds. This zinc oxide-loaded fabric (ZnO-CFs) was further changed by soaking it in a solution of ethanol and lauric acid to give it a superhydrophobic-superhydrophilic transformation surface. After that, it was soaked in a new solution of ethyl alcohol, sodium hydroxide, and water. It is possible to see ZnO adhering to the fiber surface following amendmentAnd that formed a micro-nano rough surface structure that was largely attributed to its role in the subsequent generation of hydrophobic and hydrophilic surfaces. The modified cellulose fabric’s reversible wettability retained its features, like removal efficacy and wettability stages, in spite of twenty interchange cycles between hydrophilicity and hydrophobicity. The production of sodium carboxylate, and subsequently the scissioning of chelation coupling carboxylate anion and zinc oxide generated through the procedure, was attributed to the mechanism underlying the material’s hydrophobic to hydrophilic change. The newly generated sodium carboxylate might flow from the solid-liquid channel to the liquid segment, enhancing the solid-surface free energy and result in a limited decrease in water surface tension due to the loss of low free energy alkyl chains. This causes a reversal or shift in wettability. This research not only improved the research of oil/water separation but also expanded its implementation to encompass the removal of water from oil in addition to oil from water. When the contamination is in water, the superhydrophilic/superoleophobic smart cellulose fabric can remove it. Therefore, when in its inversion condition of superhydrophobicity/superoleophilicity, it may readily remove oil from water by dipping into a solution to encourage the conversion back to the previous state, as explained before. These implementations may be adapted for the petroleum industry, where oil cleanliness is essential for its final usage, or for companies that need oil spill cleaning in enormous bodies of water [95].

Fluoroalkylsilane, synthetic polymers, and fatty acids may be used to create superhydrophobic nanocellulose, or nanocellulose can be added to certain hydrophobic films. The nanocellulose-based films are used in oil-water separation and purification, cotton and paper textile wrappings, and so on. A spray drying process was used to encapsulate a silanylated nano-cellulose-based superhydrophobic material (static CA: 164°). The chemical vapor deposition approach was used to silylate methyltrimethoxysilane (MTMS). The cellulose nanofiber length is related to the coating’s surface roughness and superhydrophobicity. Spray coating the as-synthesized material on glass and filter paper demonstrated chemical inactivity and higher durability against H_2_SO_4_, KCl solution, and deionized water. It has multiple benefits, particularly in the manufacture of rough structures [96]. By using a spray coating approach, another flexible, superhydrophobic, superoleophilic coating of nanocellulosic material was produced that could change tissue paper into an efficient oil-water segregation membrane along with an outstanding dye absorbent. Cellulose nanofiber, octadecylamine, and glutaraldehyde were used in a one-step synthesis to create the superhydrophobic material (static CA: 144°). The covered polyurethane (PU) sponge absorbed a lot of oil. As a result, all of these may be good materials for water remediation and oil spill cleanup [97].

Wang and Liu [98] created a raw cotton fiber macroporous cellulose aerogel with an adsorption capacity of 19.8–41.5 g/g by using sol–gel and freeze-drying procedures. The aerogel had excellent oil retention capabilities, and more significantly, the aerogels had exceptional superhydrophobic stability in severe conditions (alkaline, acidic, oils, and salty solutions), and can withstand many mechanical abrasions with no visible loss of superhydrophobicity. Mi and his co-workers [99] produced a superhydrophobic nanocellulose/silica fiber/Fe_3_O_4_ aerogel by direct freeze-drying and surface amendment with an adsorption capacity of 34.2–58.3. This aerogel demonstrated improved stability across a broad pH range and has various applications.

Pu et al. [100] created a Janus fabric by plasma depositing hexamethyldisilane on the cotton nonwoven surface. While the surface of the untreated fabric exhibited hydrophilicity with a CA of 0° degrees, the surface of the treated fabric exhibited superhydrophobicity (CA 150°), which was caused by the low-surface-energy substance hexamethyldisilane. Due to the nonwovens’ asymmetric wettability, they were able to demonstrate a guiding water transport operation, an excellent water vapor transportation rate of 0.19 kg m^−2^ d^−1^ with a significant air permeability of 1208 mm/s.

Zhao et al. [101] fabricated a cotton-based aerogel by means of CO_2_ supercritical drying and spray coating with an adsorption capacity of 16.0 (g/g) and a WCA of 153.0°, which is a simple, quick, and efficient superhydrophobic polymer for oil absorption, even after five cycles, and has excellent forecasts for the future in the area of oil stain removal. Dilamian and Noroozi [102] employed a simple freeze-drying procedure to create a rice straw nanocellulose aerogel with superhydrophobic and oleophobic features that may well adsorb oil and organic fluids from water with an adsorption capacity of 170 g/g. This cellulose aerogel might be an excellent adsorbent for dirty water cleanup.

The development of superhydrophobic surfaces and aerogel super absorbents has brought about potentially useful utilizations in water treatment, oil-water separation, biotechnology, medicine, and materials interfaces. Reduced transformation processes will result in a cheaper cost of the finished products, as well as environmental advantages, from reducing labor hours and material costs of the new modification phase(s).

## 7. The Applications of Hydrophobic/Superhydrophobic Structures

Adsorbents based on nanocellulose are known to enable much larger adsorption capacities than other conventional adsorbents [103]. Their primary advantages are their high surface area and the accessibility of attached functional groups. In addition, adsorption time can often be reduced because of the pore structure and shape. Nanocellulose-based products are also biodegradable, minimizing the potential for adverse environmental consequences post-operation [104]. Membranes based on nanocellulose are also believed to increase adsorption capacity in respect of impurities such as heavy metal ions and dyes.

### 7.1. Removal of Heavy Metals

Heavy metals are a major pollutant in water that renders it unfit for human consumption. It refers to metals with a density of greater than 5 g/cm^3^, which is hazardous at low concentrations. Lead, mercury, arsenic, cadmium, silver, and copper are the most common heavy metals, and the majority are toxic and associated with serious health consequences. Heavy metals can be cationic or anionic, and effective heavy metal species removal, therefore, requires appropriate modulation of the surface chemistry of the cellulose [105]. Adsorption, membrane isolation, chemical droplets, and electrochemical treatment are all used to separate heavy metal contaminants from water. Adsorption is particularly attractive because of its potentially low environmental impact and the adaptability of the technique, which enables the targeted removal of specific heavy metals.

#### 7.1.1. Heavy Metals in the Environment

In many developing countries, industrial operations and municipal waste are discharged directly into water courses, posing threats to water quality. According to Fowler [106], only 1% of global freshwater is available for everyday usage, and population growth is expected to exacerbate issues related to the scarcity of potable water in some countries. The permissible limits for heavy metals in potable water are 2 ppm (copper) [107], 5 ppm (chlorine) [108], 0.003 ppm (cadmium) [109], 0.05 ppm (chromium) [110], 0.01 ppm (lead) [111], 3 ppm (zinc) [112], and 50 ppm (nitrate) [113].

#### 7.1.2. Nanocellulose-Based Materials for Adsorption

Several studies have demonstrated the capability to extract metal ions using nanocellulose-based materials, either in pure form or after chemical modification (Table 3). Zhang and colleagues [114] synthesized TEMPO-mediated cellulose nanofibrils modified for Cu adsorption with PEI (TOCN-PEI) (II), (Figure 8). Adsorption batch experiments using aqueous Cu(II) solutions were undertaken to measure the impact of pH and adsorption time on the TOCN-PEI adsorption capability. Adsorption kinetic analyses based on the Langmuir model indicated the highest adsorption capacity of 52.32 mg/g, which was much higher than the control adsorbent. The TOCN-PEI’s reaction with Cu(II) was exothermic and enthalpic. The level of pH influenced the functional groups provided for adsorption, with a high pH being beneficial for Cu(II) adsorption. The adsorbent material was based on renewable resources and was also recyclable, with adsorption-desorption sequences revealing that the adsorption capacity could be maintained at around 33 mg/g.

The earlier advancement in adsorbents based on nanocellulose was made by Mo and colleagues [115]. Aerogel derived from wood was found to be capable of removing lead (Pb) ions in a matter of minutes and removing other metals (Mn, Cu, Zn, and Cd) at a rate of adsorption time comparable to earlier published studies [116,117,118,119,120,121,122,123]. The highest adsorption capacity for Pb ions was 571 mg/g. To offer various active sites for metal ion attachment and improved adsorption efficiency, the adsorbent was produced utilizing TOCNF, trimethylolpropanetris-(2-methyl-1-aziridine) propionate (TMPTAP), and graphene oxide (GO). The results of other adsorption studies related to nanocellulose are expressed in Table 3.

**Table 3 polymers-14-02343-t003:** Adsorbent based on nanocellulose for the removal of heavy metals.

Adsorbent Type	Targeted Heavy Metal	Production Method	Optimum Condition	Maximum Adsorption mg/g	Reference
TEMPO-oxidized CNF with PEI	Cu (II)	TEMPO oxidized cellulose nanofibers (TOCNF) grafted with Polyethylenimine (PEI)	Room Temperature/pH 5–7	52.32	[114]
TEMPO-oxidized CNF	Cu (II)	TEMPO oxidized cellulose nanofibers (TOCNF) from beech pulp fibers were prepared by using 10 mmol/g of NaClO.	Room Temperature/pH 7	135	[124]
Carboxylated cellulose nanocrystal-sodium alginate (CCN-Alg) hydrogel beads	Pb (II)	CCN-Alg adsorbent material prepared from sodium alginate in CaCl_2_ solution.	Room Temperature/pH 5.2/Contact time 180 min	338.98	[125]
Shape memory aerogels from nanofibrillated cellulose(NFC) and polyethyleneimine(PEI)	Cu(II) and Pb(II)	NPAs of NFC and PEI manufactured in an easy and green method approach via electrostatic blend.	Room Temperature/pH 2~5	175.44 & 357.14, respectively	[126]
Nanocrystalline cellulose (NCC)	Cr(III) & Cr(VI)	reinforcement using succination and amination	Room Temperature/pH 2.5~6.5	Cr(III) (94.84%) and Cr(VI) (98.33%)	[127]
Thiourea-functionalized magnetic ZnO/nanocellulose composite(TZFNC)	Pb (II)	TZFNC composite was manufactured by using a simple chemical co-precipitation technique.	Room Temperature/pH 6.5/Contact time 14.5 min	554.4	[128]
Surface functionalization of cellulose nano fibers by using methionine (Meth-CNF)	Hg (II)	CNFs extracted from rice straw were functionalized using l-methionine	Room Temperature/pH 7.8	131.86	[129]
CNC modified with NaNO_2_/NaHCO_3_	Ni (II)	sawdust-derived cellulose nanocrystals (CNC) coagulant	Room Temperature/pH 7.10	956.6	[130]
CNC	Pb (II)	Cellulose nanocrystal (CNC) from cassava peel by acid hydrolysis	Room Temperature/pH 6/Contact time 30 min	6.4	[131]
NC-PEI/GA	As (v)	Nanocellulose cross-linking polyethyleneimine and glutaraldehyde.	Room Temperature/pH 3	255.19	[132]
CNC	Pb (II)	developed from agricultural waste *Oryza sativa* husk synthesized by chemo-mechanical	Room Temperature/pH 8/Contact time 70 min	3.783	[133]
CNC/iron oxide nanorod composites	As (III) & As (V)	Acid catalyzed hydrolysis of microcrystalline cellulose	Room temperature/pH 5–9/Contact time 2 h	13.87	[134]
Poly(acryloylhydrazide)-grafted CNC Fe–Cu alloy coated CNC	Cr(VI) & Pb(II)	Atom transfer radical polymerization Oxidation-reduction method	Room temperature/pH 3	45.7 & 81.94	[135,136]

Lead and copper are the most researched heavy metals for adsorption owing to their high concentration and prevalence in wastewater. However, industrial wastewater comprises a complex combination of heavy metal ions and other contaminants, necessitating adsorbents that are engineered to target different metals. Surface modification and functionalization of nanocellulose provide routes to achieving this multifunctionality, but nanocellulose-based adsorbents also need the capacity to treat large volumes of wastewater containing several target metals. Different metals require different conditions for removal, and further research should be conducted on single and dual metal elimination, based on water-containing mixtures of metals [124,125,126,127]. In relation to nanocellulose, few publications have considered binary metal removal. For combinations comprising more than one type of metal, the most critical aspect to consider is the shape of the nanocellulose. However, the pollutant’s characteristics in the water must first be determined, whether they are cationic or anionic, in order for the adsorbent to operate appropriately. By integrating the nanocellulose-based adsorbent investigated into the membrane, improvements might be realized (adsorptive membrane). As previously stated, the major benefit of utilizing nanocellulose as an adsorbent is the simplicity with which the surface may be modified to meet application requirements [134,135,136]. Nanocellulose, CNCs, or CNFs, can be used as a membrane component or as an additive in other membranes. By integrating functional groups like amino, carboxyl, carboxylate, and silanol into the adsorption-membrane technology and surface alteration, the surface’s charge content and reactivity are changed for enhanced interaction with the pollutants present in water.

### 7.2. Removal of Dyes

Many chemicals used in textile processing are hazardous, and contamination of water sources with dyes can harm the environment, marine life, and human health. Continued interaction with dyes can cause respiratory difficulties and skin diseases, and some are mutagenic and carcinogenic. Dyeing is commonly carried out in developing countries, particularly in Asia, and dye effluent is a persistent source of water pollution and eutrophication. Dye-polluted water is challenging because commercial dyes are non-biodegradable and resistant to oxidizing agents, light, and heat. Dyes are based on aromatic compounds and can be anionic, cationic, or non-ionic [137].

Over 50% of dyes for natural fibers such as cotton are reactive dyes. Globally, about 7105 tons, consisting of 10,000 various pigments and dyes, are manufactured each year, and 1–15% of dyes are wasted in the dyeing process. Dye effluent is highly colored and has both high salt accumulation and a high chemical oxygen/biological oxygen demand ratio. Treatment of dye effluent in wastewater is therefore essential prior to discharge it into the environment [138]. Recent research studies focused on dye removal using nanocellulose adsorbents are summarized in Table 4.

Malachite green dye can be separated from an aqueous solution using a novel three-dimensional (3D), magnetic bacterial cellulose nanofiber/graphene oxide polymer aerogel (MBCNF/GOPA) that was created by Arabkhani and Afsaram [150]. MBCNF/GOPA has shown good reusability for malachite green dye removal in wastewater treatment with high adsorbent efficiency (93 percent) (Figure 9).

Recently, hybrid membranes incorporating two or further functional agents, such as self-assembled TEMPO-CNF and graphene oxide (GO) hybrid structures, have been developed for water purification. Mathew and teammates explored dye retention and the effect of GO on swelling behavior and permeance in ultrathin TEMPOCNF/GO hybrid membranes coated on a PVDF substrate. They discovered that GO hybridization minimized swelling while maintaining constant water permeance. Actual TEMPO-CNF coatings initially had high water permeance (15,000 LMHMPA), but this dropped to 6000 LMHMPA after swelling. The initial permeance was lowered to 8000 LMHMPA with 1% GO without any further decrease [156].

Functionalization is pH-dependent, resulting in a variety of maximum adsorption capacities and varying equilibrium contact durations. Compared to currently available adsorbents for dye removal, nanocellulose adsorbents exhibit equivalent or even superior performance in terms of dye adsorption capacity.

### 7.3. Filtration and Separation Media for Wastewater Treatment

Advanced membranes for wastewater purification are being developed by integrating innovative materials into porous or fibrous membrane structures. Over the last decade, there has been an increase in scientific and technological efforts focused on wastewater treatment and reuse. For example, emphasis has been placed on pressured membrane methodologies like nanofiltration (NF), microfiltration (MF), ultrafiltration (UF), forward osmosis (FO), and reverse osmosis (RO) for use in wastewater purification and marine water salt removal (Figure 10) [157]. These technologies confront three significant obstacles:A series of pre-treatment processes;Considerable energy consumption owing to high pressure;Serious membrane fouling.

The development of FO has created significant opportunities for membranes driven by osmotic pressure to address water treatment and energy issues [158]. FO membranes have a 0.4 to 1.0 nm pore size, enabling the subtraction of micropollutants, good solute refusal rate, desalination, and the elimination of heavy metals. Membranes in UF and MF are composed of copolymers of poly (vinyl chloride), poly (acrylonitrile), poly (vinylidene fluoride), polysulfone, and poly (acrylonitrile). Nitrocellulose is produced by nitrifying cellulose and is often used in dialysis and microfiltration membranes. It may also be used with cellulose acetate to make it more effective [159]. RO membranes are typically composed of cellulose acetate or polysulfone coated with polyamides [62]. NF is made of polyamide composites based on polyamide and cellulose acetate [160].

#### 7.3.1. Superhydrophobic Modified Nanocellulose for Oil/Water Separation

The exceptional wettability of textile may be categorized into two classes based on its ability to separate two distinct surface tension solution mixtures: adsorption material and filtration membrane. Individual oils or water may pass along the filtering membrane, while the other component is repelled, ensuing in a particular extraction. The absorption medium can only absorb surface oil or water, prohibiting the other component from entering. A simple and economical one-pot sonochemistry irradiation approach for fabricating a two-sided superhydrophobic textile integrated with SiO_2_ nano-particles has been devised [161]. The resulting fabric exhibited both superoleophilicity and superhydrophobicity, with a 159° WCA and a near-zero oil CA, which is suitable for oil/water isolation. The fabric may be used to capture and separate a variety of oils both above and below the water’s surface, including toluene, chloroform, and kerosene. Additionally, after 40 separation cycles, a superhydrophobic contact angle greater than 150° and an exceptional separation effectiveness greater than 94.6 percent were recorded. Furthermore, the acquired fabric exhibited stable and vigorous superhydrophobicity in the presence of strong acid, hot water, alkaline, solution of salt, and mechanical abrasion. This sturdy and robust superhydrophobic fabric surface has enormous application possibilities.

Deng and coworkers [162] demonstrated a simple and effective way of fabricating a superhydrophobic SiO_2_-TiO_2_@PDMS composite film using a sol-gel process, endowing the covered cotton-polyester fabric surface with superhydrophobicity and photocatalytic activity. The SiO_2_-TiO_2_@PDMS hybrid film can be manufactured in huge quantities and has excellent thermal solidity, equal to 400 °C. The extensive superhydrophobic fabric made in this manner was also employed as a filter for separating the oil/water combination. Additionally, the “filter cloth” demonstrated a high separation capability with a greater than 150° WCA and a sliding angle of around 8° after ten experiments.

Gao and his team [163] used electrospraying and casting to create hybrid polyvinylidene fluoride (PVDF)/SiO_2_ microspheres with a superhydrophobic covering. Separation of oil and water using gravity was discovered to be possible when filter paper with a superhydrophobic composite microsphere coating was used. Furthermore, the layered filter paper was capable of separating not only oil from drops of clean water but also corrosive droplets such as salt, acid, and alkali solution. In comparison to hybrid microspheres covered with filter paper, the detached membrane comprised of composite microspheres and ultrathin fibers showed greater efficiency in oil-water filtration. Additionally, the stretchy membrane may be employed as an adsorbent for various types of oil.

Cai and coworkers [164] presented the manufacture of extremely porous TiO_2_ microspheres using a template-assisted sol-gel technique and a template of nanocellulose aerogel. The improved spongy titanium dioxide microspheres exhibited a characteristic superhydrophobicity. Xu et al. [165] used an environmentally friendly freeze-drying method to make an absorbent three-dimensional (3D) carbon aerogel, which he then carbonized with cellulose nanofibers, graphene oxide, and poly (vinyl alcohol) resulting in aerogels with a 156° WCA and high oil penetrability.

Through freeze-drying and vacuum filtration, He and his team [166] created bacterial cellulose aerogels/silica aerogels (BCAs/SAs) with a 3D self-assembled BC skeleton as strengthening and methyltriethoxysilane sourced silica aerogels as a filler. Due to the methyl groups on the silica surface, these BCAs/SAs displayed superhydrophobicity with a contact angle of 152° and superoleophilicity. The BCAs/SAs with an exceptional capacity for oil absorption, are denoted by the quality factor Q (for organic solvents and oils). The quality factor (Q) was computed using the weight of BCAs/SAs after and before absorption using the subsequent calculation:Q = (weight after − weight before)/weight before.(5)

Sobhana et al. [167] stated that ecofriendly stearic acid hydrophobized cellulose via an interface/inorganic linker/sandwich material called layered double hydroxides because the layers have a molecular attraction for both stearic acid and cellulose. The innovative concept of utilizing the charged centers on the layered double hydroxides was successfully implemented to achieve simultaneous coupling with hydrophilic cellulose and hydrophobic stearic acid, providing the cellulose network with not only hydrophobicity but also superhydrophobicity. The oil absorption and tensile strength experiments show that stearic acid- layered double hydroxides -cell hybrid fibers are a required component of thin films, water-resistant packaging, sorbents, paper, and sanitary textiles.

Silane alteration comprises a succession of hydrolysis and distillation events, creating silanols that connect with the hydroxyl groups of nanocellulose membrane covalently. Later, hydrophobic silane linking agents are commonly utilized to transform the hydrophilic membrane into a hydrophobic nanocellulose membrane. BNC membrane improved with (tridecafluoro-1,1,2,2-tetrahydrooctyl)-trichlorosilane obtained superhydrophobicity with WCA 156° as observed by Leitch et al. [168]. They used chemical vapor deposition to produce a highly porous BNC aerogel membrane from G. medellinensis and hydrophobized it with (tridecafluoro-1,1,2,2-tetrahydrooctyl)-trichlorosilane. In membrane distillation at feed temperatures of 40 °C and 60 °C, the superhydrophobic BNC aerogel membranes were related to the commercial PVDF membrane. While the BNC aerogel membrane had twice the thickness of the productive membrane, it nevertheless obtained a high flux (22.92 ± 0.96 kg m^−2^ h^−1^) and impressive salt rejection (>99.8 percent), comparable to the PVDF membrane, due to the high porosity of the BNC aerogel membrane.

Hou et al. [169] extended the use of alkoxysilanes on BNC membranes by hydrolyzing them to modify the membrane’s superhydrophobic activities while keeping the membrane’s natural shape. Water-in-oil and oil-in-water emulsifiers including surfactants were successfully separated utilizing an amphiphilic BNC membrane with superoleophobic underwater and superhydrophobic under-oil characteristics. Oven drying retained the 3D porous structures of modified-BNC membranes in a manner comparable to that of freeze-dried BNC membranes without amendment. Cheng et al. [170] changed the BNC membrane using (3-aminopropyl) triethoxysilane prior to crosslinking with ethylenediaminetetraacetic acid because the amino group is more reactive than the hydroxyl group. Additionally, the alteration brings about a BCN membrane with tertiary amino and carboxyl groups, which facilitated Sr^2+^ adsorption to a maximum of 44.86 mg g^−1^.

Oil-water separation is a frequently utilized superhydrophobic modified fiber. Following oil-water separation, it may adsorb more heavy metals from the separated water, thus expanding its application. Table 5 summarizes the most recent published research on superhydrophobic modified nanocellulose for oil/water separation.

#### Janus Switchable Membranes

Janus switchable membranes are another form of membrane and are frequently used to separate oil from polluted water. They enable more straightforward, cost-effective filtration with a high penetration depth, as water acts as the filter medium [176]. The Janus membrane with asymmetric wettability is a new technology that has attracted attention for separating a range of oil/water blends in one isolation. Due to the contrasting wettability features on either side, the Janus membrane can operate as a switchable shield for oil/water separation. By simply changing the side exposed to the input, the Janus membrane can isolate various types of oil/water mixtures [177]. For the Janus membrane production technique, we have two layers: layer A and layer B. The asymmetric manufacturing process individually prepares layers A and B of the membrane, then combines them using successive electrospinning, sequential vacuum filtering, and other technologies. It might also be made by mixing the two ingredients in a manufacturing solution, resulting in the Janus membrane, which is attributed to the movement of insoluble ingredients inside the medium of the membrane. On the other hand, the asymmetric decoration process effectively achieves the Janus membrane, which means single-sided decoration of the as-manufactured substrate via technologies such as single-faced photo-crosslinking, single-faced photo-degradation, vapor treatment, floated deposition, single-faced coating, sequential surface modification, and so on [178,179]. The Janus cellulose membranes with contradictory (rather than asymmetric) interfacial properties on the face and back are particularly noteworthy because of their water handling and antibacterial functionalities (Figure 11) [180]. For example, one surface of the cellulose membrane is decorated with silver nanoparticles (AgNPs), while the other surface is modified with hydrophobic moieties. Thus, Janus cellulose membranes are characteristically hydrophobic on one surface and hydrophilic on the other. This membrane performed admirably, with a removal efficiency of more than 96.0 percent [181].

Xie and his colleagues created Janus membranes using regenerated cellulose membranes as the matrix consisted of biomimetic polydopamine interaction control and superhydrophobic attapulgite spraying. To create a superhydrophobic effect on the surface, superhydrophobic attapulgite was sprayed on the lower surface of the membrane using a simple spraying procedure. The polydopamine coating had a dual purpose of enhancing interfacial contacts between the organic substrate and the inorganic attapulgite and forming a super-hydrophilic layer with a hierarchical structure. The Janus membrane’s separation efficiency toward various emulsions was higher than 99 percent with a high flux rate. Due to the polydopamine layer/particle hierarchical structure and its surface similarity, the Janus membrane displayed good structural firmness even after ten cycles. Additionally, the Janus membrane demonstrated superior resource friendliness in the presence of strong acids, salts, and alkalis [182]. Agaba and colleagues also manufactured a sponge made entirely of cellulose nanofibers with asymmetric wettability on both surfaces by a freeze-drying approach followed by curing by combining two cellulose nanofiber suspensions modified variably (Figure 12). This Janus hybrid sponge is very successful in oil-water separation under gravity, with a high flux of 2687 Lm^−2^ h^−1^ for water and 3300 Lm^−2^ h^−1^ for oil and exceptional antifouling characteristics [183]. This type of sponge can be very useful for fuel purification and wastewater treatment processes.

Zhang and his colleagues developed a Janus cellulose nanocomposite membrane featuring exceptional superwetting endurance and remarkable selectivity in research. They chose cellulose paper with superhydrophilic and underwater superoleophobic characteristics as the substrate, while effectively adhering the PVDF/hydrophobic-SiO_2_ nanocomposite to the top of the substrate using high-pressure spraying electrostatic spinning technologies to obtain the superhydrophobic and superoleophilic layer. When the PVDF/hydrophobic-SiO_2_ nanocomposite layer (obverse) was on top, this membrane successfully removed microscopic oil droplets from its oil-in-water emulsions when the cellulose layer (reverse) was above. Particularly, when the gravity effect was eliminated, the membrane could also accomplish the selected one-way transmission. Even after ten repetitions of water/oil emulsion filtration, this composite membrane maintained a suitable superwetting property, separation efficiency, and flux [184].

Additionally, Janus cellulose membranes have demonstrated the ability to be reused while existing membranes demonstrate a significant reduction in efficacy and flow after only ten removal cycles. Janus cellulose membranes have the potential to be further developed for water treatment, with significant scope for upscaling and commercialization [75].

#### 7.3.2. Electrospun Membrane Filters

Cellulose acetate is among the most commonly used materials in membrane separation for water purification. Herein, water purification strategies for removing extremely toxic metal ions using functionalized cellulose acetate nanofibers are considered. Singh and Balasubramanian [185] prepared camphor soot nanospheres of 25 to 50 nm particle size and camphor soot/cellulose nanofibers by electrospinning to extract radioactive U(VI) from an aqueous solution. The membrane had a maximal adsorption capacity of 410 mg/g in 60 min. Diverse isotherm models (Langmuir, Freundlich, Temkin, and Dubinin-Radushkevich) Elovich Kinetic, and pseudo-first-order models were applied to describe the adsorption mechanism. In a research study by Zhou et al. [186] electrospun cellulose acetate/silk fibroin mixed nanofiber membranes with fibers ranging from 100 to 600 nm were reported. In an aqueous solution, the silk fibroin/cellulose acetate blend nanofiber membranes had a greater affinity for Cu(II) than pure silk fibroin or pure cellulose acetate nanofiber membranes. The mixed nanofibrous membranes with 20% CA content responded very well to Cu(II) adsorption, with a maximum milligram per gram of Cu(II) adsorbed of 22.8 mg/g. Stephen et al. [187] reported electrospinning of CA membranes, deacetylation, and oxalone-2,5-dione surface modification to produce cellulose-g-oxalone-2,5-dione. The membranes with surface modification exhibited high Pb adsorbent capacity (1.0 mmol/g) and Cd (2.91 mmol/g), relative to cellulose nanofiber. Goetz et al. [188] studied electrospun CA membranes as reinforcement for chitin or cellulose nanocrystals in the development of fully bio-based membranes (Figure 13). A hydrophilic-coated cellulose acetate structure was produced with a pore diameter of 100 Å and 85.6% porosity, suitable for a microfiltration membrane.

Electrospun membranes of CA-CNC containing cellulose nanocrystals exhibited a higher flux (13,500 Lm^−2^ h^−1^ bar^−1^) than those composed of impregnated PAN (5900 Lm^−2^ h^−1^ bar^−1^) [2]. This may be attributed to larger pore sizes, elimination of the reinforcement layer, and greater membrane hydrophilicity compared to those composed of PAN. The nanocrystal layer present in the cellulose acetate electrospun membrane appeared to enhance the hydrophilicity of the surface. In another study, Taha et al. [189] combined sol-gel and electrospinning to produce composite nanofibrous membranes. The Prepared NH(II)-functional cellulose acetate silica nanofibrous membrane (FCA/SiO_2_) had a fiber diameter of 100 to 500 nm and was utilized to remove Cr(VI) from an aqueous solution. The FCA/SiO_2_ membrane achieved an adsorption rate of 19.5 mg/g Cr(VI) by exhibiting iso-thermic Langmuir kinetics. Generally, dynamic and static adsorption studies indicated accelerated adsorption rates of under 60 min, based on high coordination efficiency, surface area, and the presence of intra- and inter-pores in the nanocellulose material. Cai et al. [190] produced composite cellulose nanofiber membranes by electrospinning cellulose acetate containing organically modified montmorillonite (MMT), followed by deacetylation. The resulting membranes showed a high capacity to adsorb Cr(VI) metal ions from an aqueous solution.

The pH of the solution, temperature, and contact time are also known to affect metal ion adsorption in nanofiber membranes. Feng et al. [191] prepared composite nanofibrous membranes composed of polyacrylonitrile/cellulose acetate (PAN/CA) by electrospinning amidoxime polyarcylonitrile/regenerated cellulose (AOPAN/RC) by integrating hydrolysis and alteration of the amidoximation. The adsorption capacities of Cd(II), Cu(II), and Fe(III) ions in the membrane were reported to be 1.13, 4.26, and 7.47 mmol/g, respectively. The adsorption equilibrium for Cd(II), Cu(II), and Fe(III) ions was achieved within 60, 20, and 5 min, respectively. Following desorption of the metal ions Cd(II), Cu(II), and Fe(III), 80% of the initial desorption rate was maintained, highlighting the potential reusability of electrospun AOPAN/RC nanofiber membranes.

Furthermore, surface-functionalized electrospun nanofibers have the potential to further enhance adsorption performance because of the selection of appropriate functionality. Celebioglu et al. [192] reported the effective removal of a polycyclic aromatic hydrocarbon (phenanthrene) by cyclodextrin-grafted CA nanofibers (CA-CD). Grafting of beta-cyclodextrin (CD) to electrospun cellulose acetate nanofibers was accomplished by click reaction, and the resulting irregular surface morphology of the fibers was thought to be indicative of surface modification. Cellulose acetate nanofibers removed 50% of the phenanthrene from the test liquid, while CD-CA nanofibers removed 64%. This was applied to the collective effect of hydrophobic interaction with the CD inclusion complexes and the high surface area of the nanofibers. Sarioglu et al. [193] produced electrospun membranes for the adsorption of ammonium from the marine environment by restriction of Acinetobacter Calcoaceticus STB1 cells in cellulose acetate nanofibers. Ammonium reductions of 72%, 98.5%, and 100% were achieved in 48 h for ammonium concentrations of 50 mg/L, 100 mg/L and 200 mg/L, respectively. At an initial ammonium concentration of 50 mg/L, the ammonium removal performance of the STB1 immobilized CA nanofibrous web was highly comparable to that of suspended bacterial cells. In contrast, bacteria-free CA webs had insignificant ammonium extraction efficiency compared to bacteria-immobilized webs. Gabru and Das [194] prepared electrospun cellulose acetate/titanium dioxide (CA/TiO_2_) membranes as adsorbents. The specific surface area was found to increase from 30.2 to 48.5 m^2^/g when the TiO_2_ nanoparticle content increased from 0 to 2.5 weight percentage, and by optimizing the membrane composition and adsorption conditions, removal efficiencies as high as 98.9% to 99.7% were achieved for lead and copper ions.

Choi et al. [195] produced a thiol-functionalized cellulose nanofiber membrane capable of adsorbing heavy metal ions from water, such as Cu(II), Cd(II), and Pb(II) via the cellulose acetate electrospinning and deacetylation route. The process involved esterifying hydroxyl groups with 3,3′-dithiodipropionic acid and cleaving the disulfide link. The adsorption characteristics of the resultant thiol-functionalized cellulose nanofiber membrane were thoroughly evaluated in relation to Cu(II), Cd(II), and Pb(II) ions. Based on the Langmuir isotherm model and the assumption that the adsorbate forms a monolayer of evenly dispersed surface adsorption energy, maximum adsorption capacities of 22.0, 45.9, and 49.0 mg/g for Pb(II), Cd(II), and Cu(II) ions, were predicted, respectively.

These findings highlight the value of surface functionality in producing high-performance adsorbent cellulose materials for effective water treatment. For heavy metal removal, functionalized electrospun nanocellulose also demonstrates adsorption efficiencies equivalent to or even superior to conventional adsorbents. Notwithstanding the high surface area, porosity, and versatility in surface functionalization options that can be achieved with nanocellulose electrospun membranes for heavy metal removal, many obstacles undoubtedly exist, particularly in relation to mass production and scalability.

#### 7.3.3. Nanopapers as Membrane Filters

Cellulose nanopaper membranes can be produced through modified papermaking processes, vacuum filtration, or solvent evaporation, resulting in sheet products in which porosity, pore size distribution, and other physical properties can be controlled. Such membranes normally comprise nanocellulose fibrils. Dense nanopapers can be produced that are capable of filtering contaminants through mechanisms of size exclusion or electrostatic interaction, depending on circumstances. Nanopaper membranes can be designed to remove metal ions, sulfates, humic acid, fluorides, phosphates, and various other organic compounds [196]. Specialty papers must be very strong, have a small distribution of particle size, and display unique optical, hydrophobic, and resistance properties. To adhere the colors to the base paper, pigments and binder chemicals are utilized [197].

The first cellulose nanopaper membranes to filter by size exclusion targeted the removal of viruses (less than 10 nm) using cellulose nanofibers or bacterial cellulose [198,199]. Subsequently, other researchers successfully harnessed this approach to remove nanoparticles, but the primary issue of nanopaper membranes is low porosity and permeability. Recently, attempts to address this for heavy metal removal have focused on modifying cellulose nanopaper production. Table 6 illustrates various methods of nanopaper production together with the target contaminants they were designed to remove. The increased accessibility to functional groups, associated with engineering higher porosity, beneficially impacts adsorption efficiency and adsorbent capacity.

Along with its chemical activity, tiny pore size, and superior mechanical qualities, cellulose nanopaper may be used to treat water. Hamed and his coworkers developed a novel membrane made up of polydopamine particles and BNC that can successfully extract metal ions and organic colors from wastewater [205]. For heavy metal ion adsorption, Zhu and co-workers developed a 1, 2, 3, 4-butanetetracarboxylic acid-modified composite membrane [206]. CNCs and polyvinyl alcohol-co-ethylene made up the composite membrane. The improved membrane was activated with NaHCO_3_ to enhance the adsorption rate. After activation by NaHCO_3_, the improved nanofiber membrane demonstrated high adsorption efficiency for single metal ions and metal ion mixtures, with an equilibrium adsorption potential of 471.55 mg g^−1^ at 15 °C. Furthermore, the composite membrane was shown to be robust after three times of repeated usage, demonstrating that the NaHCO_3_ activated membrane was a suitable material for eradicating heavy metal ions. By combining bacterial cellulose nanofibers with silica particles and self-polymerizing bio-inspired polydopamine, Wahid et al. created a superhydrophilic/superhydrophobic nanopaper-based membrane for oil/water separation [207]. With a removal efficiency of 99.9%, the nanocomposite has remarkable oil-water separation ability, indicating its promise for oily wastewater treatment in the green industry. Depending on reduced graphene oxide and bacterial nanocellulose, Jiang et al. developed an anti-biological pollution ultrafiltration membrane [208]. Under light, the ultrafiltration membrane exhibited outstanding photothermal characteristics and good bactericidal activity.

#### 7.3.4. Membrane Bioreactors

Membrane bioreactors represent a cutting-edge technology for municipal and industrial wastewater purification that harnesses nanofiltration and biological treatment processes. Membrane bioreactors provide several advantages over conventional techniques, notably reduced environmental impact, and decreased sludge and biomass preservation. Such bioreactors reduce membrane fouling through aeration, physiochemical removal, and surface enhancement. Similarly, ZnO/MnO_2_ hybrid cellulosic membranes with anisotropic interfacial properties, have been discovered to give a valuable method of water purification [209]. Lotfikatouli and coworkers evaluated the fouling characteristics of a thin-film nanofibrous composite-cellulose nanofiber coated membrane with those of industrialized use polyvinylidene fluoride membranes to filter residential wastewater composed of a membrane bioreactor capacity. In comparison to activated sludge, a thin-film nanofibrous composite-cellulose nanofiber membrane improved performance in filtering supernatant. A hydraulic wash was able to remove the cake layer that had accumulated on the membrane surface, potentially eliminating or reducing the expense of removing contaminants in practice [210]. Such asymmetric membranes are promising for potential uses such as industrial wastewater treatment, including in the fuel industry, where large quantities of wastewater are produced containing pollutants such as liquified solids, petroleum-organic constituents, and other oily compounds. For such wastewater treatment applications, functionalized cellulosic asymmetric membranes with enhanced hydrophilicity are reported to reduce fouling without sacrificing pollutant removal [181]. Membrane bioreactors for water reuse have also gotten a lot of interest in places like Singapore and China. Focusing on water recycling facilities in Spain with capacities of more than 10,000 m^3^ d^−1^, Iglesias and his team stated that the membrane bioreactor has a higher priced (approximately 30%) but comparable operating costs to sludge plants with primary treatment. The capital costs of a membrane bioreactor are found to be 10% cheaper than those of an activated sludge plant that uses advanced water treatment [211]. However, Judd [212] said that when land prices are high and treatment productivity is improved, a membrane bioreactor might be advantageous in reducing capital expenses. However, the economics of a treatment plant is still influenced by factors such as plant capacity, location, membrane lifetime, and water supply quality, among others. In the future growth of nanocellulose membranes for membrane bioreactors, prolonging the membrane life span will be a priority.

## 8. Sustainability, Challenges, and Limitations

A significant benefit of nanocellulose materials is that they are constituted entirely of renewable materials and may be readily biodegraded following disposal [213]. Over the last decade, nanocellulose and related nanocomposites have emerged that offer tremendous potential for effective water treatment. Manufacturing routes are inexpensive compared to petroleum-based nano-materials [214] and inorganic filtration membranes.

Despite these remarkable properties of cellulose, several restrictions and difficulties remain unresolved. It is acknowledged that the rigid construction of plant cell walls is due to the plentiful cellulose in nature. However, one of its drawbacks was that it relied on natural resources for production, resulting in natural disasters due to the destruction of organic material [74]. So, resources that are based on biomass are mostly a superior alternative for nanocellulose synthesis via waste valorization, as they assist in preserving the environment and conserving resources.

Nanocellulose may be efficiently surface engineered to create newer binding molecules with customized characteristics for adsorbing a variety of pollutants. It is an essential component in wastewater treatment due to its intrinsic fibrous character and outstanding mechanical characteristics, as well as its biocompatibility and low cost. The modifications of nanocellulose surfaces are growing as a present and future area to explore in the development of novel adsorbents and membranes for the removal of pollutants. Additionally, more research is required to make nano-level hybrid composites that can interact with many contaminants simultaneously.

## 9. Conclusions and Future Perspectives

The studies discussed in this review demonstrated the importance of nanocellulose materials in water pollution removal, oil/water separation via superhydrophobic treatment, and incorporation into water treatment processes. Nanocellulose, whether CNCs, CNFs, or BNC, may be transformed into membranes or added to existing membranes as an ingredient. In addition to filtering, infusion, mixing, and superhydrophobic coating, simple and economical technologies, such as interfacial polymerization, might be included in the current membrane development and fabrication. Nanocelluloses can be prepared with a strong affinity for inorganic and organic contaminants due to carboxyl, hydroxyl, carboxylate, amino, silanol, and thiol functional groups. The performance of nanocellulose membranes is additionally enhanced when combined with inorganic or organic additives, which led to the development of synergistic characteristics that nanocellulose alone could not achieve. These are all attractive characteristics for water treatment applications. Nanocellulose membranes exhibit good porosity, exceptional tensile strength, chemical resistance, antifouling capabilities, and the ability to be rendered super-hydrophilic or superhydrophobic. Superhydrophobic coatings on cellulose-based nanomaterials generally include a number of capabilities that are not possible with conventional water repellency treatments. Additionally, superhydrophobic coatings on the cellulose substrate provide a more environmentally friendly and sustainable alternative to fossil fuel-based hydrophobic polymers. It is apparent that the chemical transformation of nanocellulose membranes is important for optimizing performance and interactions with targeted contaminants. For the delivery of potable water, functionalized nanocellulose membranes have the potential to remove contaminants in less time than existing technologies. The next stage of nanocellulose-based membrane studies for water treatment will concentrate on hybrid membranes that mix nanocellulose with specific nanomaterials to increase pore size, adsorption capacity, and mechanical strength. In addition, innovative processing methods, such as 3D printing and in situ membrane functionalization, will provide a greater and more productive application of nanocellulose membranes/filters for water treatment. Its intrinsic fibrous nature, remarkable mechanical characteristics, cost-effectiveness, and biocompatibility makes it an important constituent in wastewater treatment. In addition, further research is required to build a hybrid nanocomposite of nanocellulose that can interact simultaneously with miscellaneous contaminants.

## Figures and Tables

**Figure 1 polymers-14-02343-f001:**
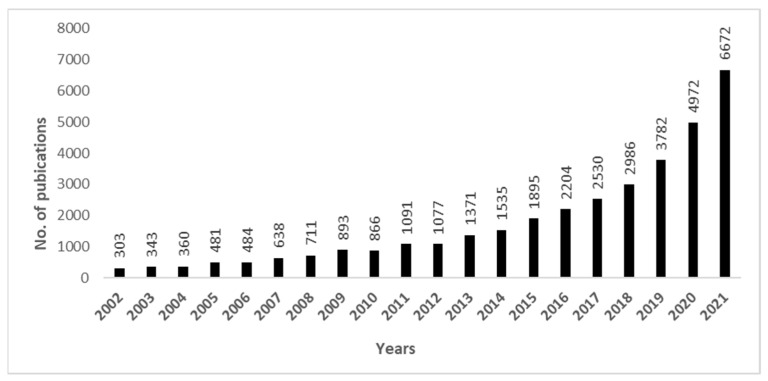
Publications on nanocellulose regarding wastewater treatment over the last twenty years (ScienceDirect database).

**Figure 2 polymers-14-02343-f002:**
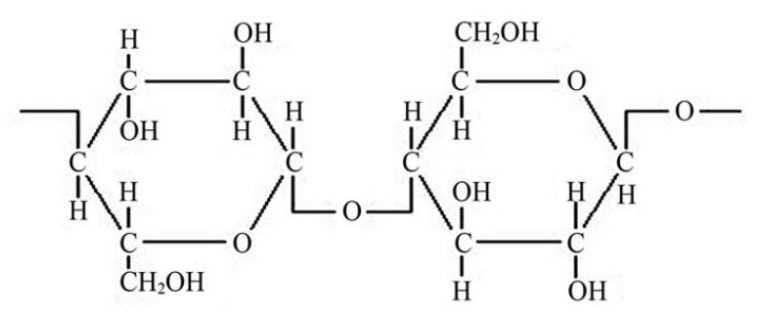
Structure of cellulose [13].

**Figure 3 polymers-14-02343-f003:**
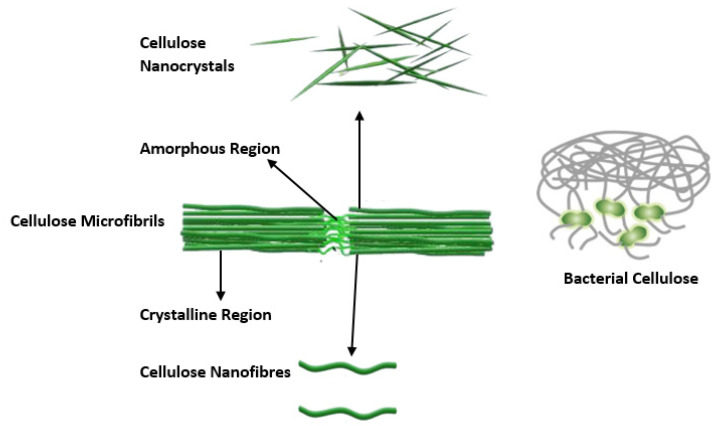
Different types of nanocellulose.

**Figure 4 polymers-14-02343-f004:**
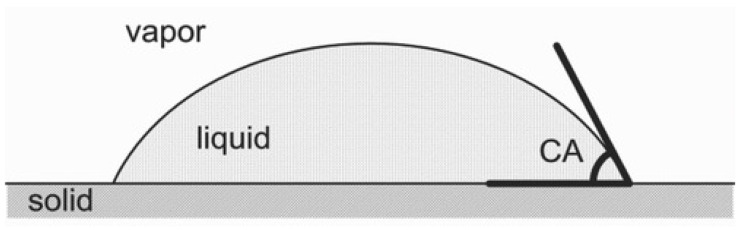
CA determination on a solid surface [10].

**Figure 5 polymers-14-02343-f005:**
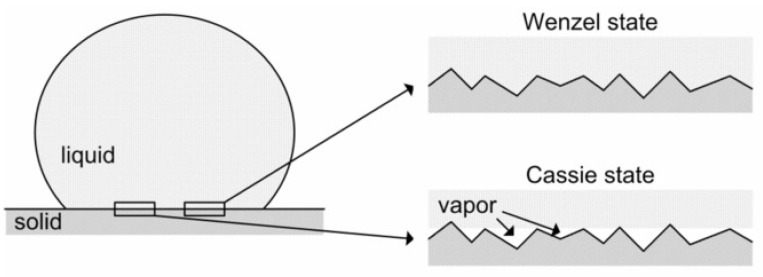
The Wenzel and Cassie wetting states are seen here on a rough surface [83,84].

**Figure 6 polymers-14-02343-f006:**
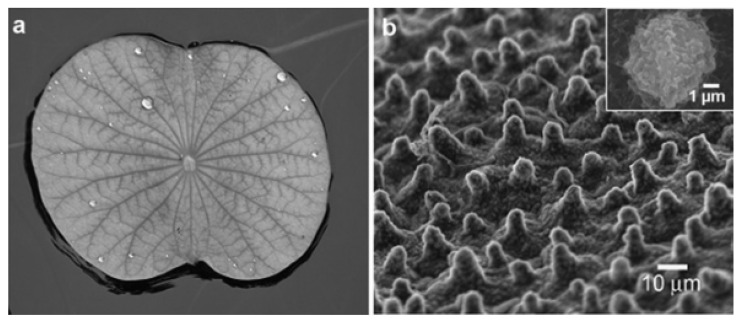
(**a**) A photograph of a lotus leaf floating on the water surface. (**b**) A SEM view of the lotus surface. The papilla structure is enhanced in the inset. Reprinted from [86]. Copyright 2005, with permission of RSC.

**Figure 7 polymers-14-02343-f007:**
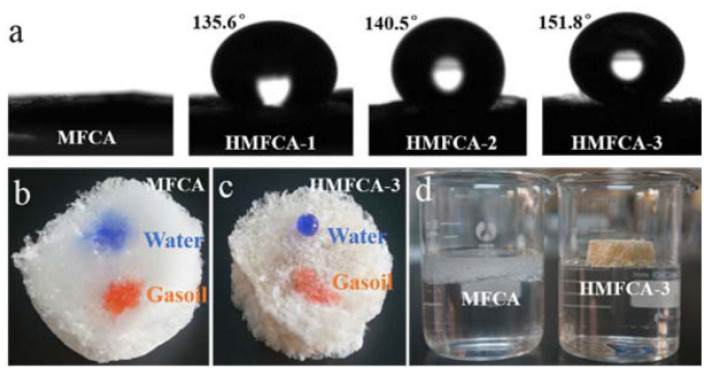
(**a**) Measuring the CA of modified and unmodified microfibrillated cellulose aerogels (MFCAs) with concentrations of 1, 2, and 3 mL of methyltriethoxysilane. (**b**,**c**) Water/oil differentiation of modified MFCA and unmodified MFCA with 3 mL of methyltriethoxysilane showing that both water and oil are absorbed in the original MFCA, but only oil is absorbed in the modified MFCA. (**d**) MFCA that has been modified floats on water, but MFCA that has not been changed sinks into water. Reproduced with permission from Zhou et al. [93]. ACS publications copyright © 2016.

**Figure 8 polymers-14-02343-f008:**
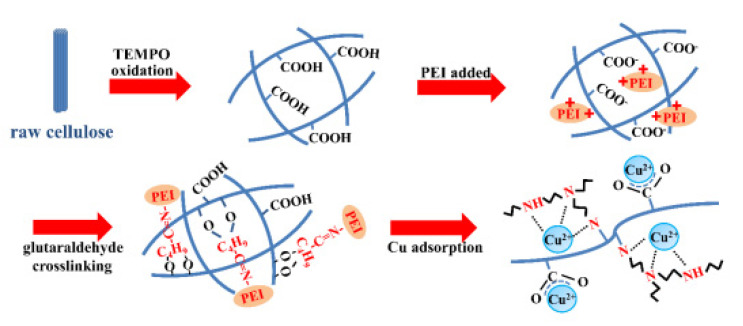
TEMPO-mediated oxidized cellulose nanofibrils improved with PEI. Reprint from [114] Copyright 2016, with permission of Elsevier.

**Figure 9 polymers-14-02343-f009:**
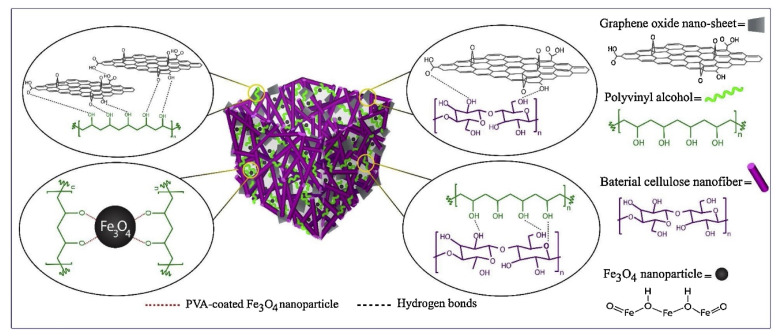
Illustration of a schematic and hypothetical interactions between graphene oxide nano-sheets, polyvinyl alcohol, bacterial cellulose nanofibers, and ferric oxide nanoparticles of magnetic bacterial cellulose nanofiber/graphene oxide polymer aerogel nanoparticles. Reprint from [150]. Copyright 2019, with permission of Elsevier.

**Figure 10 polymers-14-02343-f010:**
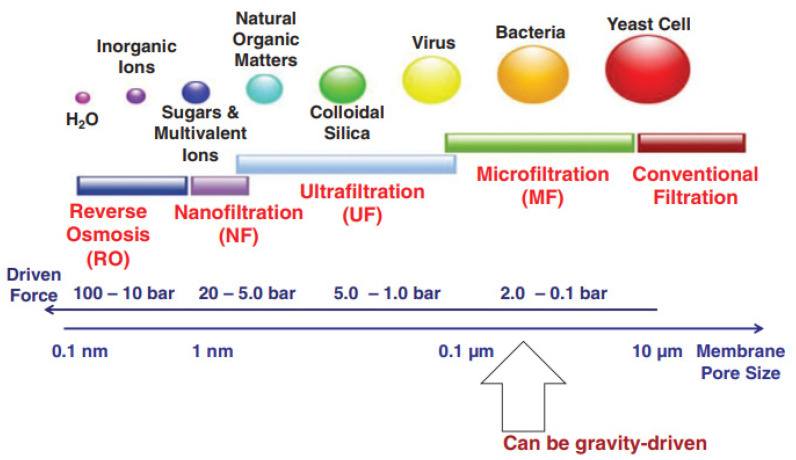
Pressure-driven membrane filtration classified according to pore size and pressure. Reprint from [157], Copyright 2020 with permission of John Wiley and Sons.

**Figure 11 polymers-14-02343-f011:**
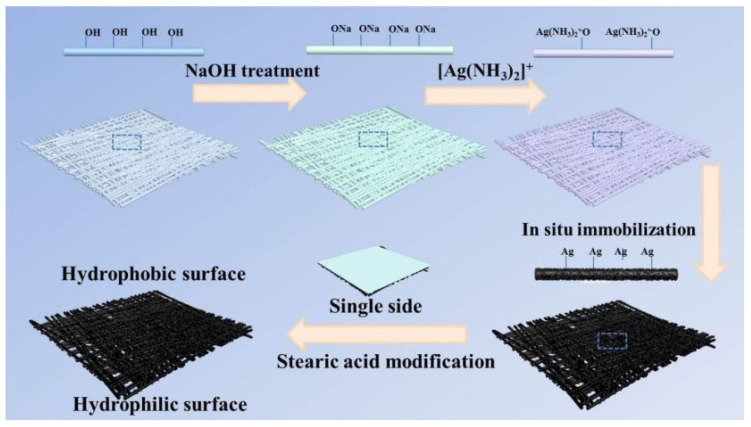
Janus cellulose membrane preparation by a facile method to enable switchable emulsion separation at the surface. Reprinted from [180]. Copyright 2019, with permission of ACS.

**Figure 12 polymers-14-02343-f012:**
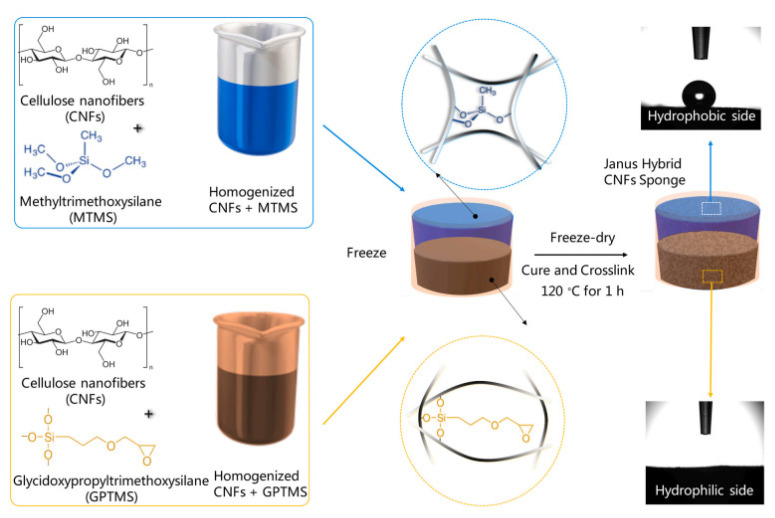
Schematic depicts the preparation methods involved in Janus Cellulose Nanofiber Sponge. Reprinted from [183]. Copyright 2021, with permission of Elsevier.

**Figure 13 polymers-14-02343-f013:**
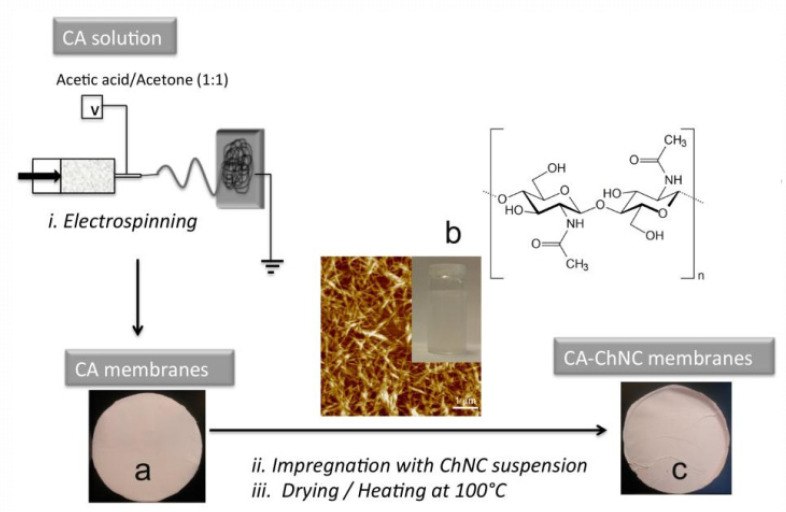
A diagram illustrating the procedures and materials used in the processing and functionalization of membranes. The process phases are as follows: (i) electrospinning of cellulose acetate (CA) mats, (ii) impregnation of CA mats, and (iii) drying and heating of impregnated mats. (**a**) The electrospun CA mat, (**b**) chitin nanocrystals (ChNC) utilized for impregnation (photo of the ChNC suspension, the AFM picture of the nanocrystals, and the chemical structure of chitin), and (**c**) the electrospun CA membrane mat formed after impregnation are exhibited. Reprinted from [188]. Copyright 2016, with permission of Elsevier.

**Table 1 polymers-14-02343-t001:** Nanocellulose characteristics data from [21].

Properties	CNFs	CNCs	BNCs
Diameter (nm)	<100	1–100	20–100
Length (nm)	Micrometer range	5–200	Micrometer range
Morphology	Long Chain	Rod/needle	Twisted ribbon
Tensile Modulus (GPa)	100	130	80–110
Tensile Strength (GPa)	0.8–1.0	8–10	1.5–1.7
Crystallinity (%)	50–65	72–80	75–80
Aspect Ratio	60–100	10–50	~50
Specific Surface Area (m^2^/g)	51	533	125

**Table 2 polymers-14-02343-t002:** Different Sources of Nanocellulose.

Source of Nanocellulose		Reference
Wood Feedstock	Hemlock	[22]
	fir, poplar, beech cherry wood	[23]
	spruce	[24]
	white cedar	[25]
	pine, aspen	[26]
	Armand pine	[27]
	American elm, red maple, paper birch	[28]
	eucalyptus wood	[29]
	oak	[30]
Agricultural Residues and Plants	coconut shell, rice husk	[31]
	Miscanthus	[32]
	rice Straw	[33]
	hazelnut shell	[24]
	switch grass	[34]
	Napier grass, jute fiber, Bermuda grass, coffee pulp	[35]
	elephant grass	[36]
	orchard grass, esparto grass and timothy grass	[37]
	oat straw	[38]
	sugarcane bagasse	[21]
	corn cobs, wheat straw, bamboo	[25]
	sisal hemp	[39]
	banana	[40]
	soybean hulls	[41]
	soybean straw	[42]
	barley straw, sweet sorghum bagasse	[43]
	cotton stalk	[44]
	pineapple leaf, sunflower stalk	[36]
	water hyacinth	[45]
Algae	algae	[46]
bacteria	bacteria	[47]
Waste	municipal solid waste	[48]

**Table 4 polymers-14-02343-t004:** Dye Removal using Nanocellulose Based Adsorbents.

Adsorbent Type	Targeted Dye	Production Method	Optimum Condition	Maximum Adsorption mg/g	Reusability	Reference
Carboxylated CNC	Methylene blue	Tempo oxidation	pH 9.0	769	-	[139]
Carboxylated CNCs	Methylene blue	Citric acid-hydrochloric acid hydrolysis	-	92.80%	-	[140]
Lignocellulosic Materials	Methylene blue	Neem oil-phenolic resin processed lignocellulosic materials	pH 2–8, time 5 min	2000	five cycles	[141]
Cellulose Nanofibers (CNFs)	Crystal violet dyes	Nonsolvent- supported approach by applying Meldrum’s acid as an esterification agent	-	3.984	-	[142]
dxe	Methylene blue	Triple-layered thin film composite nanofiltration membrane	-	96.70%	-	[143]
Microfribillated cellulose dialdehyde—chitosan composite	Congo red	Microcrystalline cellulose with high-pressure homogenization	Time 10 min	152.5	-	[144]
TEMPO-oxidized cellulose nanofibers/TiO_2_ nanocomposite	Brilliant Blue	TEMPO-oxidation accompanied by precipitation	Temp 25 °C, pH 3–8, Time 5 to 240 min	162	-	[145]
Sulphated CNCs	Auramine O	One-step ammonium persulphate oxidation	Temp 0 °C and 25 °C, Time 30 min	20	-	[146]
Electrosterically stabilized nanocryltalline cellulose	Methylene blue	Two-step oxidation by periodate and chlorite	Temp 20–60 °C, pH 1–10, Time 1 h	1250	four cycles	[147]
CNCs incorporated by Zno Nanoparticle	Methylene blue	One-pot Synthesis	Temp 25–45 °C, pH 2–10, Time 24 h	64.9	four cycles	[148]
CNC-polydopamine composite	Methylene blue	self-polymerization	Temp 25 °C, pH 2–11, Time 24 h	2066.7	four cycles	[149]
graphene oxide polymer aerogel/Magnetic BNC	Malachite Green	Combination of simple filler-loaded networks process with the aid of vacuum freeze-drying	Temp 5–45 °C, pH 2–12, Time 5–25 min	270.2	eight cycles	[150]
CNC modified by Surfactant	Congo red	Sonication	At room temperature, pH 7.5, Time 2 h	448	five cycles	[151]
Cellulose microcrystalline	Disperse yellow Dye	Surface modification	Temp 25 °C, pH 11, Time 10 min	30	-	[152]
Sodium periodate-modified nanocellulose prepared from Eichhornia crassipes	Methylene Blue	A complicated chemical process	Temp 25 °C, pH 8.0, Time 1 h	90.91	Thirteen cycles	[153]
Ethylenediamine tetra-acetic acid embedded nanocellulose	Methylene Blue	Embedment method	Temp 30 °C, pH 10	98%	-	[154]
Nanocellulose for immobilization of Laccase	malachite green and congo red	enzyme immobilization	Temp 50 °C, pH 5 for malachite green; pH 6 for congo red, Time 1 h	92% for malachite green and 62% for congo red	Eighteen cycles	[155]

**Table 5 polymers-14-02343-t005:** Oil-water separation through superhydrophobic nanocellulose-based materials.

Separator Type	Production Method	Water Contact Angle	Separation Efficiency	Reference
cellulose/poly (vinyl alcohol) composite aerogels	chemical cross-linking, freeze drying, and silanization	156.6°	98.5%	[171]
Holocellulose nanofibers	TEMPO-Mediated oxidation	149°	98.5%	[172]
Cellulose nanocrystals/polyvinyl alcohol/tetraethyl orthosilicate aerogel	Freeze-drying method	154.93° ± 4.14°	92%	[173]
Modification of wood and cotton fabric through Octadecylamine	Grafting	168.2°	97%	[174]
CNF- polydimethylsiloxane	Freez drying method	163.5°	99.9%	[175]

**Table 6 polymers-14-02343-t006:** Manufacturing Methods for Cellulose Nanopaper Production.

Cellulose Based Nanopapers	Production Method	Targeted Materials	Adsorption Capacity (mg/g)	Reference
Cellulose Nanofibers	freeze-dried	Iron	53	[200]
Ethanol phosphorylated TEMPO-oxidized cellulose nanofibrils	Cellulose nanofibrils produced from fibre sludge	Ca (II) & Mg (II)	90 & 70 respectively	[201]
TEMPO-oxidised cellulose nanofibrils	sustainable biofuels manufacturing from green algae and cyanobacteria	Ca (II)	-	[202]
Nanopaper prepared by carboxylated CNFs	-	metal ions	-	[203]
Nanopaper made from pristine fibrous NC	-	filtration of viruses and nanoparticles	-	[204]

## Data Availability

Not applicable.

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
