# Peer review of "A Review on Nanocellulose and Superhydrophobic Features for Advanced Water Treatment"

_polymers, 2022, doi:10.3390/polym14122343_

Round 1
Reviewer 1 Report
In the present review, the author has reviewed the use of nanocellulose and superhydrophobic structures for advanced water treatment. The present review is very important as we all face significant water crises across the globe. I would like to recommend this review for publication after addressing the following comments.
Comments:
1. In Figure 1 (page 3), the author could have given the source. That is, is it from Scopus or web of science or Sci finder.
2. Please correct the typo error in line number 91.
3. The author could have given a brief description of how nanocellulose is prepared.
4. Please maintain the consistency in Table 3 (page 14).
5. In Table 4, very little recent literatures are given. It would be good to review some of the recent literature as these older references are already reported in many reviews.
6. In section 6.3.1, it would be good if the author could have given one table summarizing some of the recent hydrophobic cellulose membranes for oil-water separation.
7. In section 6.3.4, very few literatures are reviewed about the membrane Bioreactors.
8. Overall, very recent literatures are not reviewed (typically from 2021 to 2022).
Author Response
Point 1: In Figure 1 (page 3), the author could have given the source. That is, is it from Scopus or web of science or Sci finder.
Response 1: They were from the ScienceDirect database, now updated in the revised manuscript as suggested.
Point 2. Please correct the typo error in line number 91.
Response 2: Corrected.
Point 3. The author could have given a brief description of how nanocellulose is prepared.
Response 2: Described briefly as suggested.
Point 4: Please maintain the consistency in Table 3 (page 14).
Response 4: Maintained as suggested.
Point 5: In Table 4, very little recent literatures are given. It would be good to review some of the recent literature as these older references are already reported in many reviews.
Response 5: Recent literature added.
Point 6: In section 6.3.1, it would be good if the author could have given one table summarizing some of the recent hydrophobic cellulose membranes for oil-water separation.
Response 6: Thank you. Table 5 is added, summarizing some recent superhydrophobic cellulose membranes for oil-water separation.
Point 7: In section 6.3.4, very few literatures are reviewed about the membrane Bioreactors.
Response 7: More literature review provided as suggested.
Point 8: Overall, very recent literatures are not reviewed (typically from 2021 to 2022).
Response 8: Recent literatures are added to expand the citation list to over 210 papers.

Reviewer 2 Report
- Manuscript title must be changed. Title must reflect the purpose of the work.
- The authors need to reorganize the current introduction, which normally consists of three parts at least: background, literature review, brief of the proposed review work. The current one is nothing but a literature review. Why is their work important comparing to existing review articles? I think this is essential to keep the interest of the reader.
- Figure 1. Publication list source? and obtained list date must be described.
- Figure 5,8 not cited in the manuscript. Line 442- Figure 4?
- All the figures must be clearly arranged. The higher resolution required.
- Many space errors/punctuation errors must be solved.
- Can authors highlight some of the n Nanocellulose materials morphology in their discussion?
- Authors are recommended to discuss the obtained results with the literature. Most of the statements are provided without bibliographic support.
- Table 5- only limited manufacturing methods reported in the literatures? Authors need to extend the literatures and discussion on it.
- Authors need to mention the outlook for the future growth of materials However, they did not provide a perspective on the future of this field. It is recommended that they include one section about their perspectives.
- Conclusions need to be improved. Authors should make it brief and consistent with their main discussion. General statements need to be removed. Authors are suggested to be more specific in their conclusions.
- The Authors are encouraged to review the form and the manuscript's English.
Author Response
Point 1: Manuscript title must be changed. Title must reflect the purpose of the work.
Response 1: Revised as “A Review on Nanocellulose and Superhydrophobic Features for Advanced Water Treatment”.
Point 2: The authors need to reorganize the current introduction, which normally consists of three parts at least: background, literature review, brief of the proposed review work. The current one is nothing but a literature review. Why is their work important comparing to existing review articles? I think this is essential to keep the interest of the reader.
Response 2: Reorganized as suggested by the reviewer. This review highlighted some recent work introducing superhydrophobicity to the nanocellulose materials and their pertinent roles in improving the functionality in water separation and/or purification.
Point 3: Figure 1. Publication list source? and obtained list date must be described.
Response 3: It was referenced from the ScienceDirect database, now also updated in the revised manuscript per your kind suggestion.
Point 4: Figure 5,8 not cited in the manuscript. Line 442- Figure 4?
Response 4: Said figures cited per suggestion.
Point 5: All the figures must be clearly arranged. The higher resolution required.
Response 5: Thank you and they have been rearranged as suggested.
Point 6: Many space errors/punctuation errors must be solved.
Response 6: Errors/punctuation errors corrected or improved.
Point 7: Can authors highlight some of the Nanocellulose materials morphology in their discussion?
Response 7: Yes, the authors have made changes so that Nanocellulose materials morphology is discussed or highlighted.
Point 8: Authors are recommended to discuss the obtained results with the literature. Most of the statements are provided without bibliographic support.
Response 8: Thank you and we have revised the text so those citations are made per iterature.
Point 9: Table 5- only limited manufacturing methods reported in the literatures? Authors need to extend the literatures and discussion on it.
Response 9: Extended the literature and discussion as suggested.
Point 10: Authors need to mention the outlook for the future growth of materials However, they did not provide a perspective on the future of this field. It is recommended that they include one section about their perspectives.
Response 10: We have attempted to provide a perspective on the benefit of superhydrophobicity.
Point 11: Conclusions need to be improved. Authors should make it brief and consistent with their main discussion. General statements need to be removed. Authors are suggested to be more specific in their conclusions.
Response 11: Well noted and improvements were made.
Point 12: The Authors are encouraged to review the form and the manuscript's English.
Response 12: Reviewed as suggested.
